# Binary Radiance Fields

**Seungjoo Shin**
GSAI, POSTECH
seungjoo.shin@postech.ac.kr

**Jaesik Park**
CSE & IPAI, Seoul National University
jaesik.park@snu.ac.kr

https://seungjooshin.github.io/BiRF

## Abstract

In this paper, we propose *binary radiance fields* (BiRF), a storage-efficient radiance field representation employing binary feature encoding in a format of either $+1$ or $-1$. This binarization strategy lets us represent the feature grid with highly compact feature encoding and a dramatic reduction in storage size. Furthermore, our 2D-3D hybrid feature grid design enhances the compactness of feature encoding as the 3D grid includes main components while 2D grids capture details. In our experiments, binary radiance field representation successfully outperforms the reconstruction performance of state-of-the-art (SOTA) storage-efficient radiance field models with lower storage allocation. In particular, our model achieves impressive results in static scene reconstruction, with a PSNR of 32.03 dB for Synthetic-NeRF scenes, 34.48 dB for Synthetic-NSVF scenes, 28.20 dB for Tanks and Temples scenes while only utilizing 0.5 MB of storage space, respectively. We hope the proposed binary radiance field representation will make radiance fields more accessible without a storage bottleneck.

## 1  Introduction

In recent years, the emergence of Neural Radiance Fields (NeRF) [1] has greatly impacted 3D scene modeling and novel-view synthesis. The methodology models a complex volumetric scene as an implicit function that maps positional and directional information of sampled points to the corresponding color and density values, enabling the rendering of photo-realistic novel views from any desired viewpoints. Subsequent advancements [2, 3, 4, 5, 6, 7, 8, 9, 10] have demonstrated their ability to reconstruct various 3D scenes using images and corresponding camera poses, which opens radiance fields as a promising approach for representing the real 3D world.

Despite the significant progress, the computational burden of utilizing large-scale multi-layer perceptrons (MLPs) remains a critical challenge, leading to a speed issue in both training and rendering radiance fields. To tackle this issue, an auxiliary explicit voxel grid has been utilized for encoding local features, denoting a voxel-based method. While the implicit radiance field representations should update all shared learnable parameters in MLPs, parametric encoding calculates only a small local part of encoded features leading to less computational cost. The voxel-based feature encoding has been implemented in various data structures, such as dense grids [11, 12], octrees [13, 14], sparse voxel grids [15], decomposed grids [16, 17, 18, 19], and hash tables [20]. These representations succeed in efficiently reducing the time required for convergence and inference. Nonetheless, explicit feature encoding methods have a significant disadvantage: their excessive storage usage. Now, we are facing a new bottleneck that restricts accessibility. Consequently, the desire for a new radiance field representation to implement a realistic 3D scene with little storage has been raised.

This paper introduces a new binary feature encoding to represent storage-efficient radiance fields with binary feature grids, referred to as *binary radiance fields* (BiRF). Here, we focus on embedding sufficient feature information restricted in binary format. We achieve this by adopting a binarization-

aware training scheme, binary feature encoding, that constraints feature encoding parameters to either $+1$ or $-1$ and update them during optimization, inspired by Binarized Neural Networks (BNNs) [21]. Accordingly, our radiance field representation can successfully reconstruct complicated 3D scenes with binary encoding parameters that can be represented using compact data resulting in a storage-efficient radiance field model. Furthermore, we extend the modern multi-resolution hash encoding architecture to a 3D voxel grid and three orthogonal 2D plane grids. This hybrid structure allows for more efficient feature capture in a more compact manner.

As a result, our radiance field model achieves superior reconstruction performance compared to prior efficient and lightweight methods. To be specific, our model attains outstanding performance in static scene reconstruction, with a PSNR of 32.03 dB for Synthetic-NeRF scenes, 34.48 dB for Synthetic-NSVF scenes, 28.20 dB for Tanks and Temples scenes while only utilizing 0.5 MB of storage space, respectively.

We summarize our contributions as follows:

- We propose *binary radiance fields* (BiRF), a binary radiance field representation that concisely encodes the binary feature values of either $+1$ or $-1$ in 2D-3D hybrid feature grid to represent storage-efficient radiance fields.

- Our binarization-aware training scheme, binary feature encoding, allows us to effectively encode the feature information with binary parameters and update these parameters during optimization.

- We demonstrate that our representation achieves superior performance despite requiring only minimal storage space, even in 0.5 MB for Synthetic-NeRF scenes.

## 2 Related Work

**Neural Radiance Fields**   Neural radiance fields (NeRF) [1] is a leading method for novel-view synthesis by reconstructing high-quality 3D scenes. To achieve scene representation, it optimizes coordinate-based multi-layer perceptrons (MLPs) to estimate the color and density values of the 3D scene via differentiable volume rendering.

Improving radiance field representations begins with embracing diverse scenarios where the scene is intricate. The sampling strategy used in the original NeRF assumes that the entire scene can fit within a bounded volume, which limits its ability to capture background elements in an unbounded scene. To address this issue, several works [2, 3] have separately modeled foreground and background by re-parameterizing the 3D space. These parameterizations have been primarily applied in unbounded $360°$ view captured scenes. Additionally, due to insufficient capacity, NeRF's lighting components have limitations in dealing with glossy surfaces. To address this challenge, transmitted and reflected radiance are optimized separately [4, 5]. Furthermore, there are approaches to extend to a dynamic domain with object movements [6, 7, 8, 22, 23].

Despite performing impressive results, it has limitations, including slow training and rendering speed. It relies solely on utilizing implicit functions for 3D scene representation, which may lead to computational inefficiencies [20].

**Radiance Fields Representations**   NeRF methods can be categorized into three types: implicit [1, 2, 3, 24, 25, 26, 27], explicit [14, 15, 18, 28], and hybrid representations [11, 12, 13, 17, 20, 23, 29, 30, 31], depending on how the approach represent the scenes.

Implicit representations extensively use neural networks to represent radiance fields, as done in the pioneering method [1]. The network has a simple structure and can render photo-realistic images with few parameters. However, they take a lot of time to converge and require significant inference time because they share the entire weight and bias parameters of the MLPs for an arbitrary input coordinate, resulting in a significant computational cost. Therefore, recent studies have proposed explicit and hybrid radiance field representations to overcome the slow speed by incorporating explicit data structures (such as 2D/3D grids or 3D points) for local feature encoding.

Explicit representations directly encode view-dependent color and opacity values with basis functions (e.g., spherical harmonics). For instance, PlenOctrees [14] bakes implicit radiance fields into an octree structure for rendering speed acceleration. Plenoxels [15] uses a sparse voxel structure, and the approach by Zhang et al. [28] utilize a point cloud. Similarly, CCNeRF [18] employs low-rank tensor

grids for 3D scene manipulation. On the other hand, hybrid representations utilize encoded local features as input for the MLPs. NSVF [13] achieves fast rendering speed thanks to the octree structure. To store local features, Point-NeRF [29] uses a point cloud, DVGO [11, 12] employs two dense voxel grids, and TensoRF [17] makes use of a factorized tensor grid. More recently, Instant-NGP [20] introduces a multi-resolution hash encoding technique that has demonstrated exceptional effectiveness in terms of convergence and rendering speed, achieving superior performance. Additionally, Zhan et al. [32] introduce a learning method of the gauge transformation in radiance fields.

While explicit and hybrid representations boost training time and rendering speed, they inevitably suffer from the critical disadvantage of large storage consumption due to excessive local features. In this study, we construct a multi-resolution 2D-3D hybrid grid by combining 2D planes and 3D grids and binarizing their encoding parameters to fully leverage their minimal information.

**Radiance Fields Compression**    Despite the acceleration of training and rendering, the explicit and the hybrid radiance fields have difficulties utilized in various applications due to their large storage. Consequently, there are several attempts to reduce the storage of the models.

For instance, PlenOctrees [14] and Plenoxels [15] filter the voxels using weight-based thresholding for leaving only the set of sparse voxels, which are sufficient to represent the scene. The distortion loss in DVGO-v2 [12] enables it to achieve better quality and greater resolution compactness. Other approaches apply bit quantization after the training. PlenOctrees [14], PeRFception [33], and Re:NeRF [34] all apply low-bit quantization of trained local features. VQAD [35] compresses the feature grid parameters into a small codebook with learned indices. Re:NeRF [34] and VQRF [36] have proposed methods for compressing existing explicit or hybrid radiance field models, including post-optimization processes. Also, Rho et al. [37] achieve high storage efficiency by applying wavelet transform on hybrid radiance field models with learnable masks.

Although the post-processing approaches above successfully reduce the storage requirements of the radiance field models, they have several disadvantages. Firstly, they require additional optimization steps for compression, which can be time-consuming. Also, their performance is bounded by the performance of the pre-trained models. In contrast, our approach performs binarization during training with the efficient 2D/3D feature grid representation. Our approach does not require any post-optimization processes and shows better rendering quality with even smaller storage space.

## 3   Preliminaries

The methodology of NeRF [1] optimizes a 5D function, as a radiance field representation, to model a continuous volumetric scene with a view-dependent effect. The implicit 5D function, which consists of MLPs, maps a 3D coordinate $\mathbf{x} \in (x, y, z)$ and a 2D viewing direction $\mathbf{d} = (\theta, \phi)$ to an emitted color $\mathbf{c} = (r, g, b)$ and a volume density $\sigma$:

$$(\mathbf{c}, \sigma) = \text{MLP}_\Theta(\mathbf{x}, \mathbf{d}). \tag{1}$$

Due to the use of an implicit function, updating all the weight and bias parameters of MLPs is necessary to train a single point. Consequently, this leads to slow convergence, requiring more than a day to optimize a scene.

For volume rendering, the colors $\{\mathbf{c}_i\}$ and densities $\{\sigma_i\}$ of sampled points along a ray $\mathbf{r}(t) = \mathbf{o} + t\mathbf{d}$ are accumulated to obtain the color of the ray:

$$\hat{C}(\mathbf{r}) = \sum_N^{i=1} T_i \alpha_i \mathbf{c}_i, \quad T_i = \prod_{j=1}^{i-1}(1 - \alpha_j), \quad \alpha_i = 1 - \exp(-\sigma_i \delta_i), \tag{2}$$

where $T_i$ and $\alpha_i$ represent accumulated transmittance and alpha of $i$-th sampled point, respectively. $\delta_i = t_{i+1} - t_i$ denotes the distance between adjacent points. Recently, an occupancy grid [20] is adopted to skip non-empty space for efficient ray sampling, leading to an advance in rendering speed.

To accelerate the rendering process, the hybrid representations [11, 17, 20] have been developed to encode local features in explicit data structures $\Phi_\theta$ (e.g., 2D/3D grid). These features are then linearly interpolated and used as inputs for small MLPs to predict colors and densities:

$$\mathbf{f} = \texttt{interp}(\mathbf{x}, \Phi_\theta), \quad (\mathbf{c}, \sigma) = \text{MLP}_\Theta(\mathbf{f}, \mathbf{d}), \tag{3}$$

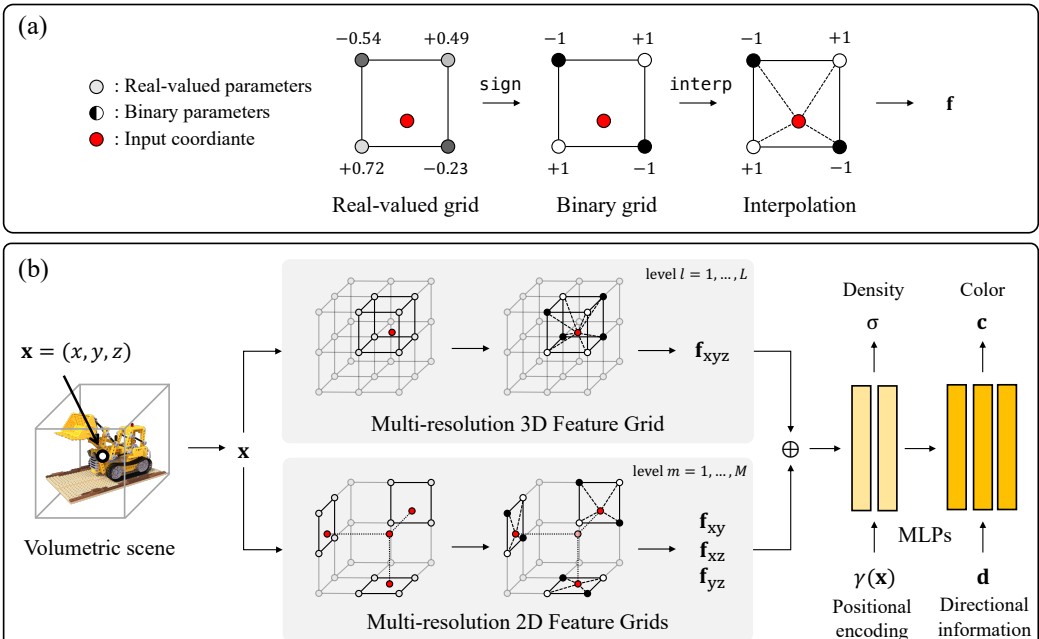

Figure 1: Illustration of overall framework: (a) binary feature encoding, and (b) binary radiance field representation. (a) Our binary feature encoding begins with applying binarization operation to the real-valued grid. Next, we linearly interpolate the binary parameters to obtain feature values. (b) Our radiance field representation comprises a 3D and three 2D feature grids. Given a 3D coordinate, the corresponding feature values are computed from each grid using binary feature encoding. The concatenated feature values are then fed as input to shallow density MLP with positional encoded coordinates. Then we can obtain the density value and embedding features that are further provided as input to shallow color MLPs to acquire color value.

where `interp` means a linear interpolation operator, and $\mathbf{f}$ denotes a interpolated feature. The choice of explicit data structure $\Phi_\theta$ significantly affects the number of learnable parameters and hence decides the total storage size of the radiance field representation. Therefore, we need to consider an efficient data structure.

## 4   Method

In this section, we introduce *binary radiance fields* that require only a small storage space by adopting binary feature grids. Fig. 1 shows the overall scheme of our radiance field reconstruction. We first introduce how to encode the binary parameters in the feature grid during optimization. Furthermore, we present our 2D-3D hybrid feature grid, which enhances the feature encoding by leveraging the strengths of both 2D plane and 3D voxel.

### 4.1   Binarization

**Binarization of learnable parameter**   The binarization procedure of real-valued variables is achieved using a deterministic binarization operator [21], the sign function, which transforms a real-valued variable into either $+1$ or $-1$:

$$\theta' = \texttt{sign}(\theta) = \begin{cases} +1 & \text{if } \theta \geq 0, \\ -1 & \text{otherwise,} \end{cases} \qquad (4)$$

where $\theta$ denotes the real-valued variable, and $\theta'$ represents binary variable.

Since the derivative of the sign function is zero almost everywhere, we cannot use traditional backpropagation to compute gradients for real-valued parameters. Instead, we use the straight-

through estimator (STE) [38], which is a simple but effective technique for backpropagating through threshold functions (e.g., the sign function):

$$\frac{\partial \mathcal{L}}{\partial \theta} = \frac{\partial \mathcal{L}}{\partial \theta'} \mathbb{1}_{|\theta| \leq 1},\tag{5}$$

where $\mathcal{L}$ is the loss function. This strategy allows us to maintain the gradient flow and ensure the differentiability of our framework. During training, we maintain and update real-valued parameters rather than directly learning binary parameters. We stop propagating gradients when $\theta$ is a large value. This also constrains the value to the range $\{-1, 1\}$ preventing the divergence of the scale.

**Binary feature encoding**   We propose binary feature encoding, a feature encoding scheme with binary parameters. Instead of real-valued parameters, binary parameters are used for encoding local features in a specific data structure, which are then employed to represent the radiance fields. While real-valued parameters require expensive floating-point data for representation, binary parameters can be expressed in a single bit (1-bit), leading to a significant reduction in storage overhead. Here, we describe how to implement this strategy in the feature grid $\Phi_\theta$. We first adopt binarization operation, described in Eq. 4, to the real-valued grid parameter $\theta$:

$$\theta' = \texttt{sign}(\theta) \quad \rightarrow \quad \Phi_{\theta'} = \texttt{sign}(\Phi_\theta),\tag{6}$$

where $\theta'$ denotes the binary grid parameter of binary feature grid $\Phi_{\theta'}$. Next, we linearly interpolate these binary parameters depending on the given coordinate $\mathbf{x}$:

$$\mathbf{f} = \texttt{interp}(\mathbf{x}, \Phi_{\theta'}) = \texttt{interp}(\mathbf{x}, \texttt{sign}(\Phi_\theta)),\tag{7}$$

where $\mathbf{x}$ is an input coordinate, and $\mathbf{f}$ denotes the encoded feature value. We utilize the STE [38] to propagate the gradients into the feature grid and update the grid parameters, as described in Eq. 5.

Now, our system is capable of training the feature encoding using binary parameters. This enables us to represent the feature grid with compact data, instead of using expensive floating-point data (16-bit or 32-bit). As a result, there is a tremendous reduction in the total storage size, making our radiance field representation storage-efficient.

## 4.2   Radiance Field Representation

For effective binary feature encoding, we design our multi-resolution feature grids with a 3D voxel grid $\Phi_{\theta_{xyz}}$, and three additional multi-resolution 2D planes $\Phi_{\theta_{xy}}, \Phi_{\theta_{xz}}, \Phi_{\theta_{yz}}$, designated to capture features along $z$-, $y$-, and $x$-axis respectively, inspired by NVP [39]. Upon this architectural design, we efficiently encode the local features and use them as inputs of MLPs to predict color and density that represent the radiance fields via volumetric rendering. Note that all feature grids are implemented using a hash encoding [20] for efficiency.

**2D-3D hybrid multi-resolution feature grid**   We intend two types of feature grids to contain the feature information in different manners. Despite the effectiveness of the 3D feature grid, the 3D grid is more severely impacted by hash collisions at higher resolutions [20], which leads to limited performance. Thus, we still need to supplement fine-grained components to improve the performance further and incorporate 2D feature grids that alleviate hash collision impact. Since the 3D feature grid encodes the main components, the 2D feature grids efficiently reinforce the feature information with fewer parameters.

**Feature evaluation**   Here, we describe the details of binary feature encoding processes to derive the local features, based on an input 3D coordinate $\mathbf{x} = (x, y, z)$.

For the 3D feature grid, we compute the local feature by tri-linearly interpolating binary parameters for each level of resolutions, and concatenating them:

$$\mathbf{f}_{xyz} = \{\texttt{interp}(\mathbf{x}, \texttt{sign}(\Phi_{\theta_{xyz}}^l))\}_{l=1}^L,\tag{8}$$

where $\mathbf{f}_{xyz}$ is the computed feature from the 3D grid, $l$ denotes the grid level and $L$ indicates the number of grid levels.

In a different way, we perform the binary feature encoding across each axis for the 2D feature grids. Firstly, we project the 3D coordinate $\mathbf{x}$ along each axis to obtain the projected 2D coordinates

$\mathbf{x_{xy}} = (x, y)$, $\mathbf{x_{xz}} = (x, z)$, and $\mathbf{x_{yz}} = (y, z)$. Next, we adopt bi-linear interpolation to extract features from these three projected 2D coordinates. Then, we acquire the feature value for each level of resolutions and concatenate them:

$$\mathbf{f_{xy}} = \{\mathtt{interp}(\mathbf{x_{xy}}, \mathtt{sign}(\Phi_{\theta_{xy}}^m))\}_{m=1}^M, \tag{9}$$

where $\mathbf{f_{xy}}$ is the computed feature from the 2D grid across $z$-axis, $m$ denotes the grid level, and $M$ indicates the number of grid levels. The feature encoding steps for other features $\mathbf{f_{xz}}$ and $\mathbf{f_{yz}}$ operate in the similar manner.

**Network architecture**    Finally, all these features are concatenated as $\mathbf{f}$ and fed into MLPs to predict the color $\mathbf{c}$ and density values $\sigma$. We utilize two MLPs each for density prediction and color prediction:

$$\mathbf{f} = \{\mathbf{f_{xyz}}, \mathbf{f_{xy}}, \mathbf{f_{xz}}, \mathbf{f_{yz}}\}, \quad (\sigma, \mathbf{e}) = \mathrm{MLP_{density}}(\gamma(\mathbf{x}), \mathbf{f}), \quad \mathbf{c} = \mathrm{MLP_{color}}(\mathbf{e}, \mathbf{d}), \tag{10}$$

where $\mathbf{e}$ presents embedded feature and $\gamma(\mathbf{x})$ is the sinusoidal positional encoding [1].

### 4.3    Loss

**Reconstruction loss**    According to the volumetric rendering process described in Sec. 3, we can render the RGB pixel values along the sampled rays and optimize them through the color and density values in Eq. 10:

$$\mathcal{L}_{\mathrm{recon}} = \sum_{\mathbf{r} \in \mathcal{R}} ||\hat{C}(\mathbf{r}) - C(\mathbf{r})||_2^2, \tag{11}$$

where $\mathbf{r}$ denotes the sampled ray encouraged by the occupancy grid. The efficient ray sampling through the occupancy grid allows us to focus on non-empty space.

**Sparsity loss**    For accelerating the rendering speed, it is important to model the volumetric scene sparsely to skip the ray sampling in the empty area using the occupancy grid. Thus, we regularize the sparsity with Cauchy loss [15, 40]:

$$\mathcal{L}_{\mathrm{sparsity}} = \sum_{i,k} \log(1 + 2\sigma(\mathbf{r}_i(t_k))^2). \tag{12}$$

The overall training loss for our radiance field model is defined as $\mathcal{L} = \mathcal{L}_{\mathrm{recon}} + \lambda_{\mathrm{sparsity}} \mathcal{L}_{\mathrm{sparsity}}$, where $\lambda_{\mathrm{sparsity}}$ is the hyper-parameter for sparsity loss. We set $\lambda_{\mathrm{sparsity}} = 2.0 \times 10^{-5}$ in this work.

## 5    Experiments

We conduct experiments on various benchmark datasets to verify the compactness of our radiance field representation. We evaluate the quantitative and qualitative results against prior works and analyze different architectural design choices.

### 5.1    Experimental Settings

**Datasets**    We adopt representative benchmark novel-view synthesis datasets. We use two synthetic datasets: the Synthetic-NeRF dataset [1] and the Synthetic-NSVF dataset [13]. Both datasets consist of eight object scenes rendered with 100 training views and 200 test views at a resolution of $800 \times 800$ pix. We also employ the Tanks and Temples dataset [41] that consists of five real-world captured scenes whose backgrounds are masked [13]. The training and test views are rendered at a resolution of $1920 \times 1080$ pix.

**Baselines**    We have three types of baseline methods, (a) data structure-based methods, (b) compression-based methods, and (c) implicit-based methods. Firstly, we compare our radiance fields with state-of-the-art radiance field representations adopting efficient data structures (DVGO [11, 12], Plenoxels [15], TensoRF [17], CCNeRF [18], Instant-NGP [20], and K-Planes [19]), which succeed to reconstruct high-quality 3D scenes with fast convergence speed. Specifically, we are interested in

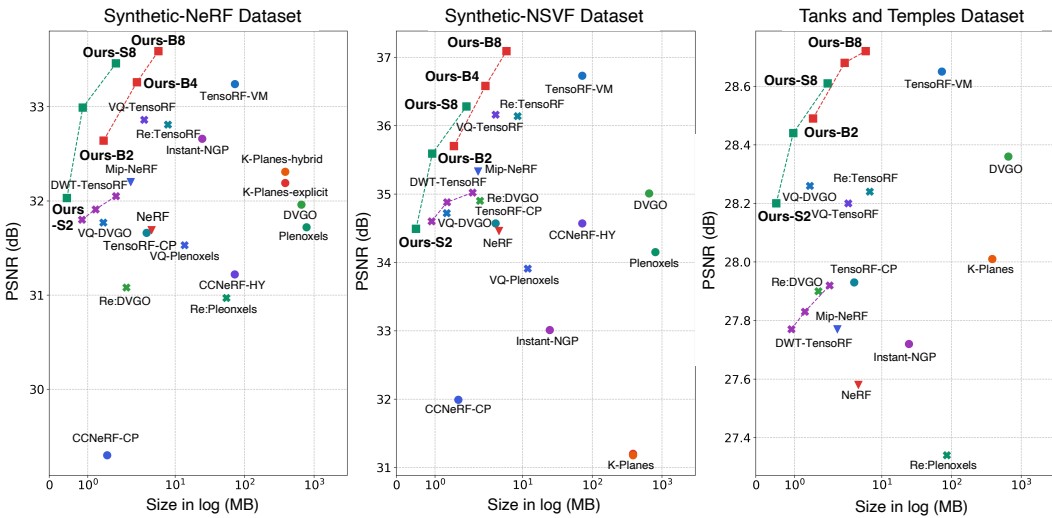

Figure 2: Comparison with baseline radiance field models on the Synthetic-NeRF dataset, Synthetic-NSVF dataset, and Tanks and Temples dataset. We utilize different dot shapes depending on the model categories: squares (■) for our models, circles (●) for data structure-based models, crosses (✖) for compression-based models, and triangles (▼) for implicit-based models.

the models that demonstrate stable performance across multiple datasets and can train within one hour when running on a conventional single GPU. Moreover, we evaluate the compression methods that require post-optimization steps (Re:NeRF [34] and VQRF [36]) and masked wavelet transform (DWT-NeRF [37]) to compress existing voxel-based models. Furthermore, we also compare with representative implicit radiance field models (NeRF [1] and mip-NeRF [24]).

**Implementation details** We implement our feature grid using hash encoding [20]. We use 16 levels of multi-resolution 3D feature grids with resolutions from 16 to 1024, while each grid includes up to $T_{3D}$ feature vectors. We also utilize four levels of multi-resolution 2D feature grids with resolutions from 64 to 512, while each grid includes up to $T_{2D}$ feature vectors. We use $\{\log_2(T_{2D}), \log_2(T_{3D})\} = \{15, 17\}$ for our small model denoting Ours-S and $\{\log_2(T_{2D}), \log_2(T_{3D})\} = \{17, 19\}$ for our base model denoting Ours-B. Our model variants are scaled by the dimension of feature vectors per level of 2, 4, and 8. For instance, **Ours-B2** denotes a base variant with feature dimension $F = 2$. We exploit two types of MLPs with 128-channel hidden layers and rectified linear unit activation on them, one for density prediction with one hidden layer and another for color prediction with two hidden layers. The total storage capacity of MLPs varies from 0.06 to 0.11 MB, depending on the input feature dimension. The spherical harmonics basis function is used to encode the directional information. Our implementation is based on Instant-NGP [20] using the occupancy grid implemented in NerfAcc [42]. We optimize all our models for 20K iterations on a single GPU (NVIDIA RTX A6000). It takes approximately 6, 9, and 14 min of average time to train scenes of the Synthetic-NeRF dataset for Ours-B2 to Ours-B8 models, respectively. We use the Adam [43] optimizer with an initial learning rate of 0.01, which we decay at 15K and 18K iterations by a factor of 0.33. Furthermore, we adopt a warm-up stage during the first 1K iterations to ensure stable optimization.

## 5.2 Comparison

We measured the storage size of each method and evaluated the reconstruction quality (PSNR, SSIM) on various datasets. Note that we present the average scores of all scenes in each dataset; scene-wise full scores are reported in the appendix. Fig. 2 summarizes quantitative evaluations of our radiance field representation, compared to baseline data structure-based, compression-based methods, and implicit-based methods.

**Data structure-based approaches** Fig. 3 shows the qualitative evaluations of our model, compared to data structure-based methods. Our method successfully demonstrates superior reconstruction performance against SOTA data structure-based methods with comparable training time and remarkably small storage size; especially Ours-S2 requires only within 0.5 MB. While baseline approaches

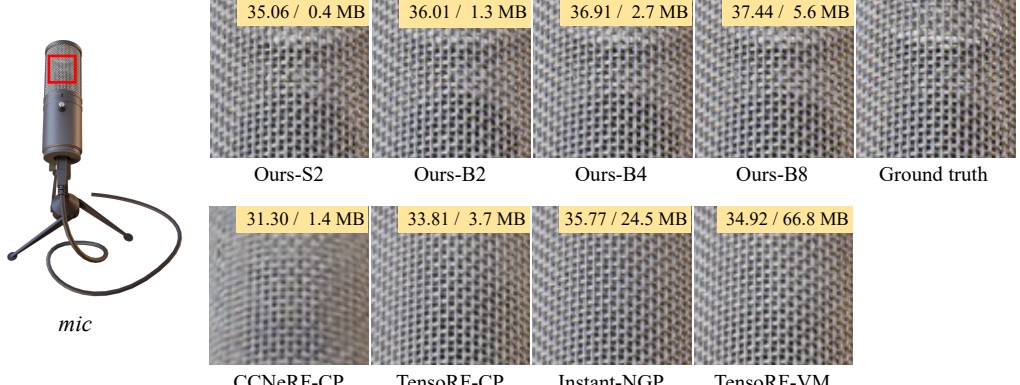

Figure 3: Qualitative comparison of reconstruction quality using the *mic* scene of the NeRF-Synthetic dataset. For each subfigure, PSNR and storage size are shown on the right upper.

require large storage space to achieve high performance, all our models are sufficient to exceed them by less than 6 MB. In particular, Ours-B2 outperforms the reconstruction quality of almost baselines with a much smaller storage size of within 1.5 MB. It takes 6.1 min to accomplish this without any temporal burden to train our model in a compact manner. Although our small models are enough to accomplish outstanding performance, we adopt larger models to attain higher quality. As a result, both Ours-B4 and Ours-B8 jump to excessive reconstruction quality, outperforming state-of-the-art models. Accordingly, the results indicate that our model contains the most storage-efficient data structure for radiance field representation, which also does not require much computational cost.

**Compression-based approaches**  We also compare our models with SOTA compression-based methods. Although these approaches make the existing data structure-based models (e.g., DVGO, Plenoxels, and TensoRF-VM) highly compact by compressing the optimized model, our binary feature encoding model outperforms highly compressed data structure-based models in terms of reconstruction performance and storage usage. Specifically, while compressed TensoRF-VM models (Re:TensoRF-High [34], VQ-TensoRF [36], and DWT-TensoRF [37]) preserve most high performance among these compression works, Ours-B4 model has superior reconstruction quality only with 2.8 MB of storage capacity, which is smaller than the compressed TensoRF-VM models. Consequently, the results verify that our binary radiance field models accomplish higher compactness than compression-based models.

**Implicit-based approaches**  In addition, we evaluate implicit representations with a few number of parameters due to the efficient structure of MLPs. Although their simple structure results in a small storage size, our models achieve significantly higher performance with a similar storage size.

**Benefits of binary radiance field**  Note that our approach differs from post quantization-based approaches [14, 33, 34] that demonstrate bounded performance from original features and necessitate additional pre-/post-processing. It is noteworthy that our method learns binary features during training without sacrificing the rendering quality. Our approach also differs from compression-based approaches [34, 36, 37] that require additional computations to compress or decompress the radiance field. Instead, our learned binary feature can be directly interpolated to render realistic scenes.

### 5.3 Ablation Study

**Feature grid design**  We analyze the impact of the feature grid designs for scene representation and verify that our 2D-3D hybrid feature grid effectively improves the reconstruction performance. We compare three architectures with similar sizes: (a) tri-plane representation, (b) voxel representation, and (c) our 2D-3D hybrid representation. Table 1 demonstrates the effectiveness of our 2D-3D hybrid representation. Compared to the tri-plane and voxel grid, our hybrid feature grid enhances the radiance field reconstruction quality with a similar number of parameters. As a result, this confirms that our architectural choice for the feature grid allows us to achieve more compact feature encoding.

Table 1: Ablation study on the feature grid design. Results are averaged over all scenes of the Synthetic-NeRF dataset. We highlight the best scores in **bold**.

| Design | # Params | PSNR↑ | SSIM↑ |
|---|---|---|---|
| Tri-plane (only 2D) | 13.7 M | 31.44 | 0.949 |
| Voxel (only 3D) | 13.6 M | 32.35 | 0.956 |
| Hybrid (2D + 3D) | 13.2 M | **32.64** | **0.958** |

Table 2: Ablation study on the use of sparsity loss. We report train time and inference speed with reconstruction quality. Results are averaged over all scenes of the Synthetic-NeRF dataset.

| | Train↓ | Inference↑ | PSNR↑ | SSIM↑ |
|---|---|---|---|---|
| Ours-B2 (w/o $\mathcal{L}_{\text{sparsity}}$) | 6.4 m | 3.3 fps | 32.69 | 0.958 |
| Ours-B2 (w/ $\mathcal{L}_{\text{sparsity}}$) | 6.1 m | 3.8 fps | 32.64 | 0.958 |
| Ours-B8 (w/o $\mathcal{L}_{\text{sparsity}}$) | 14.7 m | 2.2 fps | 33.60 | 0.964 |
| Ours-B8 (w/ $\mathcal{L}_{\text{sparsity}}$) | 13.9 m | 2.7 fps | 33.59 | 0.964 |

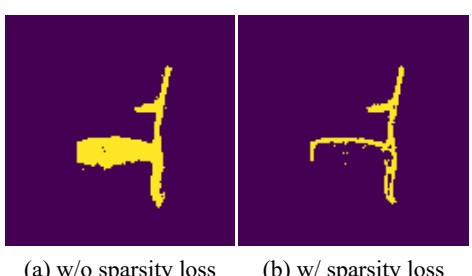

(a) w/o sparsity loss    (b) w/ sparsity loss

Figure 4: Visualization of a 2D slice of occupancy grid for *chair* scene of the Synthetic-NeRF dataset according to the use of sparsity loss $\mathcal{L}_{\text{sparsity}}$.

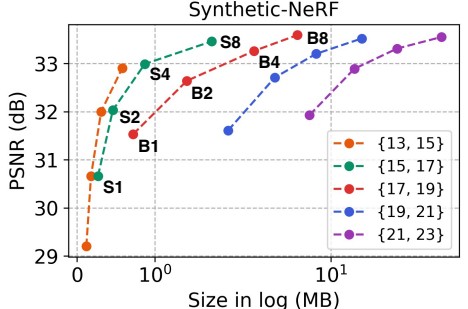

Figure 5: Ablation study on the hash table size. Results are averaged over all scenes of the Synthetic-NeRF dataset.

**Sparsity loss**   We investigate the effectiveness of sparsity loss that accelerates the rendering speed by regularizing the radiance fields more sparsely. We evaluate the train time and inference speed of our radiance field model, according to the use of sparsity loss. As shown in Table 2, we can improve the rendering time in both training and inference with a minor decrease in reconstruction quality. In particular, we can accelerate the 23% of rendering speed for Ours-B8 model by adopting sparsity loss. The visualization of the occupancy grid, including the bitmap for the empty or non-empty area, also demonstrates that our model is trained sparsely due to sparsity loss, as shown in Fig. 4.

**Hash table size**   We use hash encoding to construct our 2D/3D feature grids, which are restricted in their scale by the hash table size and the number of feature vectors per level. In other words, we can further reduce the storage size for compactness or increase the storage size to improve performance by scaling the number of feature vectors in the hash table. As shown in Fig. 5, we evaluate the storage size and the reconstruction performance of different hash table sizes $\{\log_2(T_{\text{2D}}), \log_2(T_{\text{3D}})\}$, where $T_{\text{2D}}$ and $T_{\text{3D}}$ denote the hash table size of the 2D and 3D grid, respectively.

**Computational cost for binarization**   We evaluate the training time and memory requirement of a real-valued feature grid and a binary feature grid. The binarization procedure in our model adds only a small portion of training time compared to the real-valued feature grid, as shown in Table 3. Also, there is no noticeable increase in memory usage due to binarization, as shown in Table 4. This is because we binarize only several grid parameters corresponding to a ray, not a whole grid. Furthermore, there are some cases where the binary grid converges faster or requires less memory than the real-valued grid since the sparsity of the optimized scene affects the computational cost.

Table 3: Evaluation on the training time for 20K iterations of a real-valued feature grid and a binary feature grid.

| Method | Synthetic-NeRF | | | Synthetic-NSVF | | | Tanks and Temples | | |
|---|---|---|---|---|---|---|---|---|---|
| | Ours-B2 | Ours-B4 | Ours-B8 | Ours-B2 | Ours-B4 | Ours-B8 | Ours-B2 | Ours-B4 | Ours-B8 |
| Real-valued | 6.02 min | 8.42 min | 13.31 min | 6.23 min | 8.77 min | 14.08 min | 5.93 min | 8.41 min | 13.46 min |
| Binary | 6.10 min | 8.66 min | 13.86 min | 6.22 min | 8.93 min | 14.53 min | 6.00 min | 8.59 min | 14.04 min |

Table 4: Evaluation on the memory requirement of a real-valued feature grid and a binary feature grid.

| Method | Synthetic-NeRF | | | Synthetic-NSVF | | | Tanks and Temples | | |
|---|---|---|---|---|---|---|---|---|---|
| | Ours-B2 | Ours-B4 | Ours-B8 | Ours-B2 | Ours-B4 | Ours-B8 | Ours-B2 | Ours-B4 | Ours-B8 |
| Real-valued | 5.46 GB | 6.70 GB | 10.04 GB | 5.47 GB | 6.02 GB | 10.04 GB | 6.52 GB | 7.09 GB | 11.10 GB |
| Binary | 5.45 GB | 6.58 GB | 10.06 GB | 5.47 GB | 6.30 GB | 10.02 GB | 6.51 GB | 7.92 GB | 11.10 GB |

Table 5: Quantitative evaluations of dynamic scene reconstruction using binary feature encoding. Results are averaged over all scenes of D-NeRF dataset [6] and HyperNeRF dataset [8]. † denotes that binary feature encoding is applied.

| Method | D-NeRF [6] (synthetic) | | |
|---|---|---|---|
| | Size (MB)↓ | PSNR↑ | SSIM↑ |
| TiNeuVox-S | 7.63 | 31.03 | 0.957 |
| TiNeuVox-S† | 0.59 | 28.68 | 0.938 |
| TiNeuVox-B | 47.51 | 33.02 | 0.972 |
| TiNeuVox-B† | 3.88 | 31.23 | 0.961 |

| Method | HyperNeRF [8] (real) | | |
|---|---|---|---|
| | Size (MB)↓ | PSNR↑ | MS-SSIM↑ |
| TiNeuVox-B | 47.85 | 24.20 | 0.835 |
| TiNeuVox-B† | 3.90 | 23.87 | 0.826 |

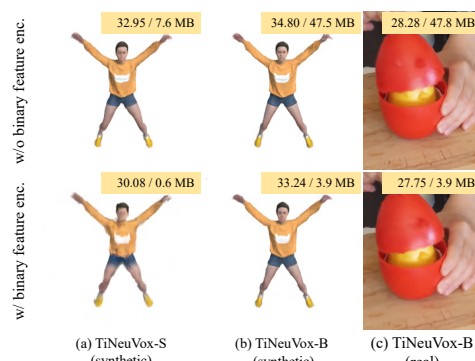

(a) TiNeuVox-S (synthetic)    (b) TiNeuVox-B (synthetic)    (c) TiNeuVox-B (real)

Figure 6: Qualitative evaluations of dynamic scene reconstruction on the use of binary feature encoding. Original (up) and binary feature encoding (bottom). PSNR and storage size are shown on the right upper.

## 5.4 Applications

**Dynamic scene reconstruction** We finally adopted our binarization strategy in dynamic scene reconstruction. We employ TiNeuVox [22], as our base model, one of the most efficient approaches, which utilizes time-aware voxel features. Instead of conventional feature encoding for the voxel grid, we apply the binary feature encoding on the time-aware voxel features and this leads to a highly compact reconstruction of dynamic scenes, even in 0.6 MB of storage space for the synthetic *jumpingjacks* scene. Table 5 and Fig. 6 demonstrate that our approach is easily applicable to various feature encoding tasks, and it enables remarkably efficient representation in terms of storage size.

## 6 Conclusion

In this work, we have introduced *binary radiance fields*, a storage-efficient radiance field representation that significantly reduces the storage capacity by adopting binary feature encoding. Our experiments have verified that our approach is the most storage-efficient representation outperforming recent data structure-based models, compression-based models, and implicit-based models in terms of reconstruction quality and storage consumption. This capability enables us to access numerous radiance fields without a storage burden and leads to the expansion of the applications, such as dynamic scenes.

**Limitations** Despite the binarization of learnable parameters, any bit-wise operations in multi-layered perception would lead to more efficient computation, requiring fewer GPU resources. We expect our approach can be boosted for the rendering by adopting the baking process.

## 7 Acknowledgments and Disclosure of Funding

The authors acknowledge the support provided by Samsung Advanced Institute of Technology (SAIT). This work was supported by IITP grants (RS-2023-00227993: Detailed 3D reconstruction for urban areas from unstructured images, No.2022-0-00290: Visual Intelligence for Space-Time Understanding and Generation based on Multi-layered Visual Common Sense, No.2019-0-01906: AI Graduate School Program at POSTECH, No.2021-0-01343: AI Graduate School Program at Seoul National University) funded by Korea government (MSIT).

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
