# OpenReview forum: "Binary Radiance Fields"
_NeurIPS.cc/2023/Conference — NeurIPS 2023 poster_

### Official Review · Reviewer_VKML · 2023-06-21

**Soundness:** 4 excellent
**Presentation:** 4 excellent
**Contribution:** 3 good
**Rating:** 8
**Confidence:** 4

**Summary:**

This paper introduces a new voxel grid radiance field representation in which the feature vectors are restricted to contain binary values.  The motivation for this is to greatly reduce the storage requirements of voxel grid radiance fields.  They use the straight-through estimator to allow backpropagation through the binary values.  The representation combines a 3D hash grid with a triplane hash grid and uses trilinear or bilinear interpolation on the binary feature vectors.  They conduct experiments on the Synthetic-NeRF, Synthetic-NSVF, and Tanks and Templs object datasets and compare against many other voxel grid radiance field methods.  They show that they are able to achieve competitive rendering quality while using an order of magnitude less storage than the best-performing competitor.

**Strengths:**

The idea is straightforward but novel to the best of my knowledge.  They are able to achieve high-quality renders with a large reduction in storage requirements.  It is surprising to see that a radiance field built only with binary feature vectors could perform so well.   They also outperform other methods in which the radiance field is compressed as a post-process.  This is probably because they are able to train the quantized model in an end-to-end manner, which is interesting to see.

There is also some innovation in the idea of combining a 3D voxel grid plus a triplane representation as usually only one or the other is used, as far as I know.

**Weaknesses:**

They are missing a reference to Variable Bitrate Neural Fields:

Takikawa, T., Evans, A., Tremblay, J., Müller, T., McGuire, M., Jacobson, A., & Fidler, S. (2022, July). Variable bitrate neural fields. In ACM SIGGRAPH 2022 Conference Proceedings (pp. 1-9).

However they include other compression-based methods in the evaluation, and VBNF did not provide results on the standard benchmark datasets, so I can understand why they didn't include it in the evaluation.

It would have been nice to see videos in the supplemental material, which would helpful for appreciating the visual quality of the results, especially for dynamic scenes.

**Questions:**

Is the model actually being stored as binary vectors during training and inference, or are you actually using 8-bit or larger integers (for example due to the way TinyCudaNN is implemented)?   I wondered if it is actually feasible to represent binary vectors in this way using PyTorch, or if that is only a hypothetical currently.

**Limitations:**

The technical limitations appear reasonable; there is no mention of broader social impacts.

---

> ### Author Rebuttal · Authors · 2023-08-09
>
> We appreciate your thoughtful comments. Following your comments, we will cite the missing related work in the manuscript (Q1, Q2 in global response) and supplement per-scene videos for the results in the supplementary material (Q2). Also, we have elaborately explained several questionable parts of the detailed implementation of binary data (Q3). The detailed responses to your comments are as follows.
>
> ---
>
> **Q1. However they include other compression-based methods in the evaluation, and VBNF did not provide results on the standard benchmark datasets, so I can understand why they didn't include it in the evaluation.**
>
> Thank you for understanding the exclusion of VBNF (referred to as VQAD) [1] as our baseline model in evaluation. As the reviewer mentioned, it does not contain results for standard benchmark datasets that we use. Please refer to the Q2 in the global response for a more detailed comment.
>
> ---
>
> **Q2. Addition of video for checking visual quality of the results, especially for dynamic scenes.**
>
> Thank you for suggesting the supplement to the video. We will add per-scene videos for each dataset, including dynamic scenes. Furthermore, we plan to show these videos on the project page when available.
>
> ---
>
> **Q3. Is the model actually being stored as binary vectors during training and inference, or are you actually using 8-bit or larger integers (for example due to the way TinyCudaNN is implemented)? I wondered if it is actually feasible to represent binary vectors in this way using PyTorch, or if that is only a hypothetical currently.**
>
> During training and inference, the grid parameters are stored in floating-point data type since we implement our feature grid using tiny-cuda-nn [2] that only supports floating-point data type (16-bit or 32-bit).
> Also, we currently cannot implement actual binary vectors because most common frameworks, including PyTorch, do not support data types of a single byte. Thus, it is only available to use 8-bit (1-byte) or larger data types to represent binary parameters. Accordingly, we still need to cast n (≥8) binary parameters and store them in the n-bit data type. Our optimized binary parameters are also stored in 8-bit integers due to such limitations.
> As we mentioned in Sec.6 in the manuscript, we acknowledge that the un-optimized implementation is a limitation of our work. We expect it to be refined by implementing an optimal binary feature grid.
>
> ---
>
> **Reference**
>
> [1] Takikawa et al., Variable bitrate neural fields, SIGGRAPH 2022
>
> [2] Müller, Tiny cuda neural networks, https://github.com/NVlabs/tiny-cuda-nn

---

> > ### Comment · Reviewer_VKML · 2023-08-14
> >
> > I have read over all of the reviews and the authors' responses.  Thanks to their authors for their explanations and additional experimental results.  I think the additional results and explanations strengthen the paper further, and I am still supportive of acceptance.

---

### Official Review · Reviewer_fKiM · 2023-06-27

**Soundness:** 3 good
**Presentation:** 4 excellent
**Contribution:** 3 good
**Rating:** 7
**Confidence:** 5

**Summary:**

This paper proposes a new representation, binary radiance fields (BiRF), for memory-efficient novel view synthesis tasks. The representation is inspired by the binary neural network. BiRF is built upon Instant-NGP. The critical component of this representation is the binarization of real-valued feature grids, such that the resulting feature grids can store bitwise feature grids, which highly reduces the storage of NeRF models. BiRF also enhances the 3D voxel grid with three 2D plane grids, where the 2D features are incorporated to alleviate the hash collision. The training loss composes of the RGB loss and the sparsity loss. Experiments are conducted on the NeRF-synthetic dataset and NSVF dataset. Compared with state-of-the-art NeRF methods (data structure-based and compression-based methods), BiRF can obtain comparable or even higher reconstruction quality while requiring less storage. The ablation study also shows the effectiveness of introducing the 2D planes and sparsity loss.

**Strengths:**

The paper is well written. The idea is simple but very effective and easy to implement. The insight of combining 3D feature grids and 2D plane grids is really cool. Overall, the method proposed in this paper is critical to the computer vision community (both the industrial and academia). For example, memory storage is an issue if we are reconstructing very large-scale scenes; there is also a demand for deploying NeRF models to mobile devices.

**Weaknesses:**

The paper also mentioned a limitation is that it requires a longer time to train BiRF compared to its Instant-NGP counterpart, due to the binarization operation on real-valued feature grids. Moreover, I think the memory requirement during training can be higher than the non-binarized version since BiRF needs to maintain the temporal real-valued feature grids in addition to the binary feature grids.

**Questions:**

The reconstruction quality of BiRF outperforms other methods by a large margin in terms of PSNR. But the reason seems not very clear since BiRF replaced the real-valued feature grids with the binary feature grids. In other words, the performance should drop (slightly) compared to the non-binary version (Instant-NGP). I think the performance gain is from the incorporation of 2D plane features. However, the authors did not provide ablations on with and without the binarization of their network architecture. I would definitely rate my score if the authors provide that.


**Limitations:**

Some minor typos are that l and L are not explained for Eq.(8), though it is obvious they're the number of grid levels. (the same issue for Eq (9)).

---

> ### Author Rebuttal · Authors · 2023-08-09
>
> We appreciate your thoughtful comments. Following your comments, we have additionally performed more experiments on the binary feature encoding for training time (Q1), memory requirement (Q2), and reconstruction quality (Q3). The detailed responses to your comments are as follows. We have compared a real-valued (or un-binarized) feature grid and a binary feature grid. Please note that “real-valued” refers to a real-valued feature grid that does not apply binary feature encoding, and “binary” refers to a binary feature grid proposed in this work.  The detailed responses to your comments are as follows.
>
> ---
>
> **Q1. Longer training time of the proposed BiRF during training.**
>
> We have evaluated the training time for 20K iterations of a real-valued feature grid and a binary feature grid, as shown in the below table. Although the increase in training time is the limitation of our model, it is not critical since we utilize a simple binarization -- a sign function. Also, the binarization procedure takes only a small portion of the whole pipeline so that such simple computation does not significantly impact the total training time. Furthermore, there are some cases where the binary grid converges faster than the real-valued grid since the sparsity of the optimized scene affects the training time, as described in Sec. 5.3 in the manuscript.
>
> **Experiment 1) Evaluation of the training time for 20K iterations of a real-valued feature grid and a binary feature grid.**
>
> ||Ours-F1|Ours-F2|Ours-F4|Ours-F8|
> |:---|:---:|:---:|:---:|:---:|
> |***Synthetic-NeRF***|||||
> |Real-valued|5.15 min|6.02 min|8.42 min|13.31 min|
> |Binary|5.12 min|6.10 min|8.66 min|13.86 min|
> |***Synthetic-NSVF***|||||
> |Real-valued|5.31 min|6.23 min|8.77 min|14.08 min|
> |Binary|5.17 min|6.22 min|8.93 min|14.53 min|
> |***Tanks and Temples***|||||
> |Real-valued|5.04 min|5.93 min|8.41 min|13.46 min|
> |Binary|5.01 min|6.00 min|8.59 min|14.04 min|
>
> ---
>
> **Q2. Higher memory requirement of the proposed BiRF during training.**
>
> The memory requirement during training can be higher than the non-binarized version since BiRF needs to maintain the temporal real-valued feature grids and the binary feature grids.
>
> We have evaluated the memory usage of a real-valued feature grid and a binary feature grid. As shown in the below table, there is *no noticeable increase in memory usage* due to binarization.
> This is because we binarize only several grid parameters corresponding to a ray, *not a whole grid*. Thus, the additional memory usage from the binarization is not as much as we need to concern.
> Also, the number of samples per ray is an important factor for memory usage. As we use an occupancy grid for efficient ray sampling, the memory usage is also affected by the sparsity of the optimized scene. This leads to less memory requirement of the binary grid for several scenes, despite the additional computation of binary feature encoding.
>
> **Experiment 2) Evaluation of the memory requirement of a real-valued feature grid and a binary feature grid.**
>
> ||Ours-F1|Ours-F2|Ours-F4|Ours-F8|
> |:---|:---:|:---:|:---:|:---:|
> |***Synthetic-NeRF***|||||
> |Real-valued|4.39 GB|5.46 GB|6.70 GB|10.04 GB|
> |Binary|4.40 GB|5.45 GB|6.58 GB|10.06 GB|
> |***Synthetic-NSVF***|||||
> |Real-valued|4.36 GB|5.47 GB|6.02 GB|10.04 GB|
> |Binary|4.37 GB|5.47 GB|6.30 GB|10.02 GB|
> |***Tanks and Temples***|||||
> |Real-valued|5.42 GB|6.52 GB|7.09 GB|11.10 GB|
> |Binary|5.41 GB|6.51 GB|7.92 GB|11.10 GB|
>
> ---
>
> **Q3. Can you provide ablations on the binary feature encoding? [Fig.4 in the attached PDF]**
>
> We have additionally performed ablations on the binary feature encoding. Specifically, we have compared the rendering quality of a real-valued feature grid denoting “real-valued” and a binary feature grid denoting “binary.” As shown in Fig.4 in the attached PDF, there is a drop in rendering quality when we use the binary grid rather than the real-valued grid with the same resolution setting. Nonetheless, we need to focus on the fact that the binary grid achieves a highly compact model size with an impressive compression rate. Also, please note that our binarized grid with multi-bit is comparable to the similar or smaller storage size of a real-valued grid (e.g., F1 of real-valued vs. F8 of binary).

---

> > ### Comment · Reviewer_fKiM · 2023-08-18
> > **Thanks for the rebuttal**
> >
> > The authors have answered all of my questions. I decided to mantain my rating for this paper.

---

### Official Review · Reviewer_ptCH · 2023-07-02

**Soundness:** 3 good
**Presentation:** 3 good
**Contribution:** 3 good
**Rating:** 6
**Confidence:** 5

**Summary:**

The paper proposes a novel approach called binary radiance fields (BiRF) for efficient storage and representation of radiance fields. BiRF utilizes binary feature encoding, where local features are encoded using binary parameters of +1 or -1. This compact encoding significantly reduces storage size and computational costs. The authors introduce a binarization-aware training scheme and extend the multi-resolution hash encoding architecture to a 3D voxel grid and orthogonal 2D plane grids.
The contributions of the paper are 1. The introduction of binary radiance fields (BiRF) as a storage-efficient representation that encodes features using binary parameters. 2. A binarization-aware training scheme that effectively captures feature information and updates binary parameters during optimization. 3. The demonstration of superior reconstruction performance with minimal storage space usage, achieving impressive results in static scene reconstruction.

**Strengths:**

1. The paper introduces a novel approach called binary radiance fields (BiRF) for representing radiance fields using binary feature encoding. This idea of employing binary parameters to represent local features in radiance fields is innovative and distinguishes it from traditional methods. The application of binarization-aware training and the extension of multi-resolution hash encoding to a hybrid structure further contribute to the originality of the approach.

2. The paper demonstrates high-quality research through rigorous experimentation and evaluation. The proposed BiRF representation outperforms state-of-the-art methods in terms of reconstruction performance while utilizing significantly less storage space. The experiments conducted on various scene datasets provide compelling evidence of the effectiveness and efficiency of the proposed approach.

3. The paper is well-written and effectively communicates the concepts and methodologies to the readers. The authors provide clear explanations of the key ideas, including the binary feature encoding, binarization-aware training scheme, and the hybrid structure of the feature grid. The organization of the paper enables easy comprehension of the research objectives, methodology, and experimental results.

4. The paper's contributions have significant implications for the field of radiance fields and 3D scene modeling. By introducing the BiRF representation, the authors address the critical challenge of storage efficiency in radiance field models, which can greatly impact practical applications. The superior reconstruction performance achieved by BiRF, coupled with its minimal storage requirements, opens up new possibilities for real-world implementation and broader accessibility of radiance fields.

**Weaknesses:**

While the paper demonstrates several strengths, there are also a few areas where it could be improved:

Experimental Evaluation: While the paper presents compelling results by reporting the model size and psnr, the paper lacks a quantitative evaluation and comparison of the inference speed with other relevant methods such as TensoRF and Instant-NGP. To what extent does the hybrid 3D and 2D feature grid architecture impact the inference speed & training speed of the proposed model? As the backward gradient to the grid is estimated & approximated, will the binary design affect the convergence speed?
What is more, it seems the reported training speed of TensoRF is slower than the original paper.

Results: In Figure 2, I find the the results of K-Planes on Synthetic NSVF dataset are significant worse than other methods, certain analysis of the inferior results should be discussed. It is crucial to consider and provide information about the training time and inference speed, as these factors play a significant role in assessing the effectiveness and practicality of the proposed NeRF model.

Hash Collision: In original Instant-NGP, the hash collision is explicit handled as the largest gradients—those most relevant to the loss function—will dominate the optimization, the multi-scale will also alleviate it as collision is statistically unlikely to occur simultaneously at every level for a given pair of points. For this proposed BiRF, how about the situation of hash collision compared with instant-ngp? With binary code, it seems the multi-scale design will be less effective to prevent hash collision.

Binarization of learnable parameter: The binarization with straight-through-estimator is a special case vector quantization or discrete representation, which has been explored in some recent NeRF research [1,2]. The paper could benefit from a more thorough discussion of these related works. Besides, there appears to be some duplication of content between lines 165-168 and lines 148-151. Streamlining these sections would improve the clarity of the paper.

Reference:

[1] Variable bitrate neural fields

[2] General neural gauge fields




**Questions:**

1. I would suggest the author to provide an evaluation of the inference speed with other relevant methods such as TensoRF and Instant-NGP, and analyze the effect of 3D&2D feature grid and estimated gradient.
2. It would be helpful to discuss the potential reasons for this performance gap of K-plans and provide an analysis of the inferior results.
3. How does the proposed BiRF model handle hash collision compared to Instant-NGP?
4. It would be beneficial to provide a more thorough discussion of these related works and explain the specific connections and differences between the proposed BiRF approach and the existing literature.
5. Streamlining the mentioned sections to enhance the clarity of the paper.


**Limitations:**

The limitation is discussed in this work.
There is no concern of potential societal impact.

---

> ### Author Rebuttal · Authors · 2023-08-09
>
> We sincerely appreciate your thoughtful comments. Following your comments, we have performed more experiments on inference speed (Q1 in global response, Q1) and convergence speed (Q1, Q2). Also, we have clearly explained several questionable parts (Q3, Q4) and the strategy for hash collision (Q5). Furthermore, we will cite the additional related works (Q2 in global response). The detailed responses to your comments are as follows.
>
> ---
>
> **Q1. Impact of 2D-3D hybrid feature grid architecture on the inference and training speed.**
>
> We have further evaluated the inference speed (FPS) and training time (min) of different grid designs: (a) tri-plane representation, (b) voxel representation, and (c) our 2D-3D hybrid representation. Note that the number of parameters of the three representations is similar.
> As shown in the below table, the inference and training speed of 2D-3D hybrid representation is faster than tri-plane (2D), while slower than voxel grid (3D). This is because the tri-plane representation needs more computations to get local feature values. Specifically, local features in the voxel grid are computed by linear interpolation of 9 nearest points. In contrast, the tri-plane needs to perform linear interpolation of 4 nearest points for three 2D planes each, a total of 12 points. Thus, a more computational cost of the tri-plane leads to a slower speed. In conclusion, the training and rendering speed of our 2D-3D hybrid representation is positioned in the middle of the tri-plane and voxel representation as ours combines them.
>
> **Experiment 1) Comparison of inference speed (up) and training time (bottom) according to the different grid design.**
>
> ||Synthetic-NeRF|Synthetic-NSVF|Tanks and Temples|
> |:---|:---:|:---:|:---:|
> |***Inference speed$\uparrow$ (fps)***|
> |Tri-plane (only 2D)|3.56|3.98|0.71|
> |Voxel (only 3D)    |4.46|4.79|0.82|
> |Tri-plane (only 2D)|3.83|4.64|0.91|
> |***Train time$\downarrow$  (min)***|
> |Tri-plane (only 2D)|8.69|8.34|8.30|
> |Voxel (only 3D)    |5.37|5.51|5.28|
> |Tri-plane (only 2D)|6.10|6.22|6.00|
>
> ---
>
> **Q2. Impact of binary feature encoding on the convergence speed. [Fig. 3 in the attached PDF]**
>
> Following your suggestion, we have explored the impact of binary feature encoding on the convergence speed by comparing a real-valued feature grid (w/o binary feature encoding) and a binary feature grid (w/ binary feature encoding). As shown in Fig. 3 in the attached PDF, we have evaluated the rendering quality of these two different feature grids according to the training time. We have validated five test views for each scene to reduce time-consuming. In the early stages, we observe that the binary grid converges faster than the real-valued grid. However, the real-valued grid outperforms within 1 min, and both grids reach the fully converged performance at a similar time. Therefore, there is no degradation of convergence speed due to binarization, despite the different rendering quality.
>
> ---
>
> **Q3. Slow training speed of TensoRF reported in the manuscript compared to the original paper.**
>
> We have followed the official code of TensoRF [1] with the default configuration. Different environmental settings, such as hardware might cause the inconsistency. We use a single NVIDIA RTX A6000, while the authors of TensoRF use a single NVIDIA Tesla V100. As we have performed all experiments, including baselines, in the same environmental setting, the fairness of our comparison is ensured.
>
> **Experiment 3) Comparison of training time for TensoRF models.**
>
> ||TensoRF-CP-384|TensoRF-VM-192|
> |:---|:---:|:---:|
> |Original paper|25.2 min|17.4 min|
> |Ours|24.7 min|21.5 min|
>
> ---
>
> **Q4. Performance gap of K-Planes on the Synthetic NSVF dataset in Fig. 2.**
>
> Thank you for your detailed concern about the results. As shown in Table 7 in the appendix, the score of K-Planes for the *Lifestyle* scene is noticeably lower than other models. In the *Lifestyle* scene, we have found a severe artifact that causes low performance and does not disappear by changing the random seed number. This result might yield confusion, as pointed out, so we will add comments for the failure case. If reviewers indicate that the K-Planes results for the Synthetic-NSVF dataset are not reasonable, we are willing to exclude them.
>
> ---
>
> **Q5. How about the situation of hash collision compared with instant-ngp? [Fig. 5 in the attached PDF]**
>
> The multi-scale design also helps to prevent the hash collision. Still, the key idea mitigating the hash collision of our model is the 2D-3D hybrid grid representation alleviating the hash collision explicitly.
>
> In Instant-NGP [2], the main idea of a hash grid is to represent a large number of grid points with only a small size of the hash table. Thus, we cannot avoid the hash collision, which means more than two grid points are mapped to the same index in the hash table. To reduce the frequency of hash collisions, we need to decrease the number of grid points we represent. Therefore, we consider a 2D hash grid of $O(N^2)$ whose number of grid points is less than a 3D hash grid of $O(N^3)$ to alleviate the hash collision, where N is the resolution of the grid.
>
> As shown in Fig. 5 in the attached PDF, we have quantified the average frequencies of hash collision according to different grid designs, 2D grid and 3D grid. There are severe hash collisions in the 3D grid at higher resolutions, while the 2D grid yields fewer hash collisions even at higher resolutions. Therefore, we combine 2D hash grids, which have less hash collision frequency, with a 3D hash grid to improve the restricted performance due to severe hash collisions. As a result, this 2D-3D feature grid can obtain higher rendering quality, as shown in Table 1 of the manuscript.
>
> ---
>
> **Reference**
>
> [1] Chen et al., Tensorf: Tensorial radiance fields, ECCV 2022
>
> [2] Müller et al., Instant neural graphics primitives with a multiresolution hash encoding, SIGGRAPH 2022

---

> > ### Comment · Reviewer_ptCH · 2023-08-18
> >
> > The response clearly solves my concerns. Thus, I improve my final rating from 5 to 6.

---

### Official Review · Reviewer_Bcw2 · 2023-07-07

**Soundness:** 3 good
**Presentation:** 3 good
**Contribution:** 2 fair
**Rating:** 4
**Confidence:** 5

**Summary:**

This paper proposes a novel binary radiance fields (BiRF) which binarized the feature encoding to save memory usage of NeRF. In the experiments, the binary radiance field representation demonstrates superior reconstruction performance compared to state-of-the-art efficient radiance field models, all while requiring lower storage allocation. Notably, the proposed model achieves remarkable results in reconstructing static scenes, achieving PSNR values of 31.53 dB for Synthetic-NeRF scenes, 34.26 dB for Synthetic-NSVF scenes, and 28.02 dB for Tanks and Temples scenes. These impressive outcomes are attained using minimal storage space, with only 0.7 MB, 0.8 MB, and 0.8 MB utilized, respectively. The intention behind introducing the binary radiance field representation is to eliminate storage bottlenecks and make radiance fields more accessible for various applications.






**Strengths:**

1. This paper is well-organized
2. Experiments are convincing and extensive.

**Weaknesses:**

1. Only binarizing the feature encoding to save the memory usage is limited, since the acceleration rate is also import in network quantization and the inference of NeRF.

2. The proposed binarization of learnable parameter is limited of novelty. The analysis or discuss about bottleneck of NeRF quantization or binarization is lack, which is crucial.

**Questions:**

See weakness.

**Limitations:**

Yes.

---

> ### Author Rebuttal · Authors · 2023-08-09
>
> We appreciate your comments. We have found that your concerns are mainly from two sources: the limited contribution of binary feature encoding (Q1) and the analysis of the bottleneck of NeRF quantization (Q2). Thus, we focus on addressing these concerns in this rebuttal. The detailed responses to your comments are as follows.
>
> ---
>
> **Q1. Only binarizing the feature encoding to save the memory usage is limited, since the acceleration rate is also import in network quantization and the inference of NeRF.**
>
> We agree that acceleration is one of the critical issues for more practical NeRF. However, we aim to address the issue of large storage size, which restricts the accessibility of recent radiance fields. Since this storage problem yields a bottleneck in NeRF advancement, several works have also been concerned with the seriousness and handled it with post-quantization [1, 2, 3] and post-optimization [3, 4]. Nonetheless, there is a limitation, as we described in Sec. 2 in the manuscript, so we propose our BiRF to solve the large storage problem by adopting binary feature encoding.
>
> Although acceleration is not our main target, we have considered it. First, we employ a hash grid of instant-NGP [5], one of the SOTA methods having both fast convergence and inference time, as the base grid implementation of the proposed BiRF. Moreover, we regularize the sparsity of the scene to achieve further improvement in rendering speed, as described in Sec. 4.3 & Sec. 5.3 in the manuscript. There is room for further improvement of inference speed since our binary feature encoding can be easily applied in other feature encoding models, as described in Sec. 5.4 in the manuscript.
>
> ---
>
> **Q2. The proposed binarization of learnable parameter is limited of novelty. The analysis or discuss about bottleneck of NeRF quantization or binarization is lack, which is crucial. [Fig. 6 in the attached PDF]**
>
> Although we introduced several works (PlenOctrees [1], PeRFception [2], Re:NeRF [3]) that employ quantization in L107-108, it might be insufficient to fully explain the previous NeRF quantization approaches.
>
> Previous methods [1, 2, 3] apply 8-bit quantization for their learned feature values after optimization to reduce the final storage size of the radiance field model. However, there is a severe degradation of rendering quality when we try to quantize the feature encoding parameters to a lower bit, as shown in the below table and Fig. 6 in the attached PDF. This is because post-quantization yields information loss, existing as a bottleneck of NeRF quantization.
>
> To quantify the loss from the post-quantization, we have evaluated the rendering quality of the quantized models (1-, 2-, 3-, 4-, 8-bit) using post-quantization methods following previous methods [1, 2, 3]. To be specific, we first optimize the models using a binary feature grid (w/ binary feature encoding) denoting “ours (1-bit)” and a real-valued feature grid (w/o binary feature encoding) denoting “base (16-bit)”. After then, we quantize the optimized real-valued feature grid from 16-bit to n-bit, denoting “post-quant (n-bit)”. As shown in Fig. 6 in the attached PDF and the below table, there is a significant drop in rendering quality as the parameters are quantized to a lower bit. Binarization (or 1-bit quantization) especially leads to severe information loss, so we can not observe the texture of the target scene. In contrast, our model still has high rendering quality despite using 1-bit data since we update the binary parameters during optimization. Therefore, we expect that our binarization strategy solves the existing bottleneck of NeRF quantization and can be considered an important contribution to our work.
>
>
> **Experiment 1) Comparison of the reconstruction performance according to the post-quantization. "Post-quant. (n-bit)" denotes post-quantization to n-bit data.**
>
> ||PSNR|SSIM|LPIPS|
> |:---|:---:|:---:|:---:|
> |Post-quant. (1-bit)|16.85|0.797|0.219|
> |Post-quant. (2-bit)|17.85|0.933|0.078|
> |Post-quant. (3-bit)|25.33|0.958|0.048|
> |Post-quant. (4-bit)|31.61|0.962|0.041|
> |Post-quant. (8-bit)|33.68|0.963|0.039|
> |Ours (1-bit)|32.64|0.959|0.049|
> |Base (16-bit)|33.69|0.963|0.039|
>
> ---
>
> **Reference**
>
> [1] Yu et al., Plenoctrees for real-time rendering of neural radiance fields, ICCV 2021
>
> [2] Jeong et al., PerfCeption: perception using radiance fields, NeurIPS 2022
>
> [3] Deng et al., Compressing explicit voxel grid representations: fast nerfs become also small, WACV 2023
>
> [4] Li et al., Compressing volumetric radiance fields to 1 mb, CVPR 2023

---

> > ### Comment · Area_Chair_yZVz · 2023-08-18
> >
> > Dear Reviewer,
> >
> > Since the discussion with authors is closing soon, could you please go over the reviews and rebuttals, and respond to the content of the authors response with a message to the authors (you can post with one message summarizing all such reviews). It is important that authors receive a reply to their rebuttals, as they have tried to address comments raised by the reviewers.
> >
> > -AC

---

### Official Review · Reviewer_EsYg · 2023-07-09

**Soundness:** 3 good
**Presentation:** 3 good
**Contribution:** 2 fair
**Rating:** 6
**Confidence:** 4

**Summary:**

In this paper, the authors are proposing BiRF (BInary Radiance Fields), a storage-efficient representation for neural radiance fields. The technique relies on a hybrid representation that leverages explicit feature grids (both one 3D and three 2D, each at multiple resolutions) combined with density and color MLPs. To achieve high storage efficiency, the feature grids are binarized following the technique from Binarized Neural Networks. The reconstruction loss includes a sparsity inducing loss (similarly to SNeRG and Plenoxels).

The main technical contributions of the paper are this specific storage-efficient representation of neural radiance fields with a matching training scheme and an array of comparisons against other methods.

The authors demonstrate results through a number of quantitative evaluations on the Synthetic-NeRF, Synthetic-NSVF and Tanks&Temples datasets and against multiple baselines (fast ones: DVGO, Plenoxels, TensoRF, CCNeRF, Instant-NGP and K-Planes and compact ones: Re:NeRF, VQRF):
- the proposed representation is indeed very storage efficient, with < 1 MB at acceptable quality,
- it generally delivers quality reconstructions with lower storage requirements compared to either efficient or compressed alternatives,
- training time remains reasonable (only behind Instant-NGP, DVGO and Plenoxels depending on the operating point).

Lastly, the storage efficiency of the technique is showcased to be relevant for an application on dynamic scenes.

**Strengths:**

Storage efficiency is an important aspect of making neural radiance fields more practical for concrete applications. While there are several neural radiance fields approaches focusing on trading rendering (and training) speed at the expense of storage efficiency and other approaches tackling storage efficiency separately by compressing a trained representation, the proposed approach addresses storage efficiency directly without necessitating a post-processing step after training and also without sacrificing quality.

The approach offers competitive operating points in terms of quality v.s. storage, while offering reasonable training time.

The method is conceptually simple and the paper does a solid job at presenting it, demonstrating its value through both quantitative and qualitative experiments, including a large number a baselines to compare against. The ablations are also thorough.

**Weaknesses:**

Novelty is limited as this is essentially an application of Binarized Neural Networks to neural radiance fields approaches.

As quality is one of the claims of the approach, a comparison against a few non-compressed and non-optimized baselines (e.g. original NeRF, mip-NeRF) would have made sense.

The approach is exploring with good results another trade-off compared to the efficiency-oriented techniques that sacrifice storage. However, the only mention of rendering speed (at test time) are in the ablation on the sparsity loss and in the limitations section. The apparently unoptimized implementation is another weak point of the submission and a comparison of rendering speed, especially against the considered baselines would have also made sense.

Minor corrections:
- l.104 Pleoxels -> Plenoxels
- l.135 consider the -> consider an

**Questions:**

Following the above, would the authors be able to include a thorough comparison of rendering speed against the considered baselines?

Binarized Neural Networks, which this paper is following, was using in some of their experiments stochastic binarization (as a form of regularization), is this something that has been considered?

Table 3 in the supplementary material suggests increasing, have the authors tried going beyond {2^19,2^21}? Are the results plateau-ing?

---

> ### Author Rebuttal · Authors · 2023-08-09
>
> We sincerely appreciate your thoughtful comments. We have additionally performed more experiments following your comments: inference speed (Q1 in global response), more baselines (Q1), and extended ablation study (Q3). Also, we positively consider adopting stochastic binarization for further improvement of binary feature encoding (Q2). The detailed responses to your comments are as below.
>
> ---
>
> **Q1. Evaluation on original NeRF and mip-NeRF.**
>
> We provide additional results for original NeRF [1] and mip-NeRF [2] in our comparison. We have followed the default architectural setting of the original paper using the code implemented in NeRF-Factory [3]. We optimize both NeRF and mip-NeRF for 300K iterations with a batch size of 4,096 on a single NVIDIA RTX A6000. It takes about 24 hours to converge for a single scene.
> As shown in the below table, both NeRF and mip-NeRF show less reconstruction quality with a larger storage size compared to Ours-F2. Despite the lightweight network architecture of implicit NeRFs, we have proved that our model (Ours-F2) demonstrates higher performance in a more compact storage size than NeRF and mip-NeRF.
>
> **Experiment 1) Quantitative evaluation of original NeRF and mip-NeRF.**
>
> (We reported the averaged score of each dataset.)
>
> ||Size (MB)|PSNR|SSIM|LPIPS|
> |:---|:---:|:---:|:---:|:---:|
> |***Synthetic-NeRF***|
> |NeRF [1]    |4.6|31.69|0.951|0.065|
> |Mip-NeRF [2]|2.3|32.20|0.955|0.062|
> |**Ours-F2**|**1.4**|**32.64**|**0.959**|**0.049**|
> |
> |***Synthetic-NSVF***|
> |NeRF [1]    |4.6|34.46|0.967|0.044|
> |Mip-NeRF [2]|6.1|35.33|0.971|0.039|
> |**Ours-F2**|**1.5**|**35.40**|**0.976**|**0.024**|
> |
> |***Tanks and Temples***|
> |NeRF [1]    |4.6|27.58|0.902|0.171|
> |Mip-NeRF [2]|6.1|27.77|0.901|0.171|
> |**Ours-F2**|**1.5**|**28.44**|**0.916**|**0.122**|
>
> ---
>
> **Q2. Consideration of stochastic binarization.**
>
> Thank you for suggesting stochastic binarization [4]. We agree that stochastic binarization might yield an interesting regularization effect. Despite the strengths, our current approach does not consider stochastic binarization. The reason is that we wanted to optimize a 3D scene as fast as possible. Thus, it is beneficial to use simple and effective deterministic binarization rather than stochastic binarization that generates random numbers.
>
> We have compared the performance of stochastic binarization and deterministic binarization (ours), as shown in the below table. As a result, applying stochastic strategy directly tends to be unstable during optimization. Compared to deterministic binarization, stochastic binarization takes a longer time and shows lower rendering quality. However, properly combining stochastic strategy with deterministic binarization may lead to better performance. Consequently, we consider employing regularization via stochastic binarization for further improvement in our future work.
>
> **Experiment 2) Comparison of binarization using Ours-F2.**
>
> (We reported the averaged score of each dataset.)
>
> ||Train time (min)|PSNR|SSIM|LPIPS|
> |:---|:---:|:---:|:---:|:---:|
> |***Synthetic-NeRF***|
> |Stochastic   |14.16|29.81|0.924|0.093|
> |Deterministic| 6.10|32.64|0.959|0.049|
> |***Synthetic-NSVF***|
> |Stochastic   |11.53|27.64|0.913|0.108|
> |Deterministic| 6.22|35.40|0.976|0.024|
> |***Tanks and Temples***|
> |Stochastic   |9.13|27.77|0.888|0.174|
> |Deterministic|6.00|28.44|0.916|0.122|
>
> ---
>
> **Q3. Further ablation on the hash table size described in Table 3 in the supplementary material. [Fig. 2 in the attached PDF]**
>
> Thank you for noticing the extensive ablation on the hash table size {$\log{T_{2D}}$, $\log{T_{3D}}$}, where $T_{2D}$ and $T_{3D}$ denote the hash table size of the 2D and 3D grid, respectively. We have performed additional ablations on the smaller and larger hash table size, {13, 15} and {21, 23}. As shown in Fig. 2 in the attached PDF and the below table, it is no longer beneficial to use a larger hash table {21, 23} because there is only a slight improvement in reconstruction quality while the storage size significantly increases. In contrast, a significant drop in rendering quality is observed when we use a smaller size of the hash table {13, 15} due to severe hash collisions. The results show that hash collision does not considerably affect the performance from the size {17, 19}, so we choose {17, 19} as the default setting of our model.
>
> **Experiment 3) Additional ablation on the hash table size {$\log{T_{2D}}$, $\log{T_{3D}}$}. Results are averaged over all scenes of the Synthetic-NeRF dataset.**
>
> ||Ours-F1||Ours-F2||Ours-F4||Ours-F8||
> |:---|:---:|:---:|:---:|:---:|:---:|:---:|:---:|:---:|
> ||**Size (MB)**|**PSNR**|**Size (MB)**|**PSNR**|**Size (MB)**|**PSNR**|**Size (MB)**|**PSNR**
> |{13, 15}|0.12|29.20|0.18|30.66|0.31|32.00|0.58|32.90|
> |{15, 17}|0.27|30.66|0.46|32.03|0.87|32.99|1.73|33.46|
> |{17, 19}|0.72|31.53|1.14|32.64|2.83|33.26|5.76|33.59|
> |{19, 21}|1.94|31.61|3.99|32.71|7.84|33.20|16.61|33.51|
> |{21, 23}|7.05|31.93|14.70|32.89|29.65|33.31|61.35|33.55|
>
> ---
>
> **Reference**
>
> [1] Mildenhall et al., Nerf: Representing scenes as neural radiance fields for view synthesis, ECCV 2020
>
> [2] Barron et al., Mip-nerf: A multiscale representation for anti-aliasing neural radiance fields, ICCV 2021
>
> [3] Kakao Brain, Nerf-factory: an awesome pytorch nerf collection, https://github.com/kakaobrain/nerf-factory
>
> [4] Hubara et al., Binarized neural networks, NeurIPS 2016

---

> > ### Comment · Area_Chair_yZVz · 2023-08-18
> >
> > Dear Reviewer,
> >
> > Since the discussion with authors is closing soon, could you please go over the reviews and rebuttals, and respond to the content of the authors response with a message to the authors (you can post with one message summarizing all such reviews). It is important that authors receive a reply to their rebuttals, as they have tried to address comments raised by the reviewers.
> >
> > -AC

---

> > ### Comment · Reviewer_EsYg · 2023-08-20
> >
> > I have read the rebuttal and have been following the discussions. I want to thank the authors for the overall rebuttal and I do appreciate them taking the time to answer all questions from the reviewers including mine.
> >
> > All the remarks I had have been addressed. The comparison against unoptimized baselines is convincing and can help support the quality claim. At this point, the remaining weaknesses are the novelty and the implementation whose efficiency could have been pushed further. I am thus upgrading my overall rating from borderline accept to weak accept.

---

### Author Rebuttal · Authors · 2023-08-09

**Global response**

Dear reviewers,

We thank all reviewers for their insightful feedback. As highlighted by reviewers, our paper proposes concise (EsYg, fKiM, VKML) and innovative (ptCH, VKML) ideas and is well-written (Bcw2, ptCH, fKiM). Also, we are delighted that the reviewers thoroughly agree on the importance of storage efficiency for NeRF models (EsYg, ptCH, fKiM). As additional verification and explanation are required by the reviewers, we have tried to respond to all these valuable concerns within a limited rebuttal period. Typos and additional results will be updated in our revision. Please refer to the attached PDF containing figures for global response and response to each reviewer.

---

**Shared response for inference speed**

**Q1. Evaluation on the inference speed (EsYg, ptCH). [Fig.1 in the attached PDF]**

We have evaluated the inference speed of our models and baselines, as shown in Fig.1 in the attached PDF and the below table. Despite the highly compact storage size, our models demonstrate a comparable inference speed, especially Ours-F2 shows 3.83 fps, 4.64 fps, and 0.82 fps for each dataset, respectively. However, several models (DVGO [1], Plenoxels [2], Instant-NGP [3]) still show faster rendering speed, so we need to consider adopting an acceleration method, such as an efficient ray sampling or binary operation, as our future work. There are no significant obstacles since our binarization strategy can be applied to any feature encoding method as a plug-in. Additionally, the slow speed of Ours-F1 is from the inefficiency of atomic half-precision accumulation implemented in tiny-cuda-nn [4], a library for our hash grid implementation.

**Experiment 1) Comparison of inference speed (FPS).**

We reported the average FPS of each dataset.
||Synthetic-NeRF|Synthetic-NSVF|Tanks and Temples|
|:---|:---:|:---:|:---:|
|DVGO|5.90|6.69|1.29|
|Plenoxels|10.35|7.52|2.31|
|TensoRF-CP|0.71|0.77|0.16|
|TensoRF-VM|1.20|1.25|0.28|
|CCNeRF-CP|1.16|1.22|0.24|
|CCNeRF-HY|1.01|1.07|0.22|
|Instant-NGP|3.90|4.23|1.36|
|K-Planes-explicit|0.88|0.91|0.24|
|K-Planes-hybrid|0.75|0.91|0.29|
|Ours-F1|3.70|4.45|0.80|
|Ours-F2|3.83|4.64|0.82|
|Ours-F4|3.41|4.13|0.64|
|Ours-F8|2.72|3.45|0.45|

---

**Shared response for related work**

**Q2. Discussion on additional related work (ptCH, VKML).**

Thank you for noticing more relevant works. We will cite the mentioned papers [5, 6] in related work. In particular, VQAD [5] successfully compresses the feature grid parameters into a small codebook with learned indices rather than using a hash function. Nonetheless, expensive data is still needed for each grid point to represent the learned indices of the codebook, while ours needs only binary information. Moreover, as commented by reviewer VKML, it does not directly fit for representative NeRF benchmark we use. Specifically, it may require depth information for pre-processing, but the datasets we used do not contain depth maps. Please understand excluding VQAD in our comparison, despite its strengths.

---

**Shared response for the manuscript**

**Q3. Modification of the manuscript on typos, duplicated contents, and missing explanations for variables (EsYg, ptCH, fKiM).**

Thank you for noticing the parts that need to be modified in the manuscript. We will correct the minor typos in L104 & L135 and add explanations for variables in L191 & L197. Moreover, we have found that L148-151 and L165-168 contain similar contents, as reviewer ptCH mentioned. We will modify to briefly introduce the use of STE in L165-168 because it is already described in L148-151. When it is available, we will update all of them in the revised manuscript.

---

**Reference**

[1] Sun et al., Direct voxel grid optimization: Super-fast convergence for radiance fields reconstruction, CVPR 2022

[2] Fridovich et al., Plenoxels: Radiance fields without neural networks, CVPR 2022

[3] Müller et al., Instant neural graphics primitives with a multiresolution hash encoding, SIGGRAPH 2022

[4] Müller, Tiny cuda neural networks, https://github.com/NVlabs/tiny-cuda-nn

[5] Takikawa et al., Variable bitrate neural fields, SIGGRAPH 2022

[6] Zhan et al., General neural gauge fields, ICLR 2023

---

### Decision · Program_Chairs · 2023-09-21

**Decision:**

Accept (poster)

**Comment:**

The final scores of this paper are SA, A, WA, WA, BR. Although one reviewer express a concern that proposed binarization of learnable parameter is limited of novelty, all other reviewers think that this paper has made important technical contributions. The rebuttal has done an excellent job in providing additional experiments and explanations. The AC has checked the paper, the review comments and the discussions, and agree with majority of the reviewers. Please incorporate all the additional materials in rebuttal into final version of this paper.